# Automated Detection of Rice Bakanae Disease via Drone Imagery

**DOI:** 10.3390/s23010032

**Published:** 2022-12-20

**Authors:** Donghoon Kim, Sunghwan Jeong, Byoungjun Kim, Seo-jeong Kim, Heegon Kim, Sooho Jeong, Ga-yun Yun, Kee-Yeun Kim, Keunho Park

**Affiliations:** 1IT Application Research Center, Korea Electronics Technology Institute, Jeonju 54853, Republic of Korea; 2Horticultural Research Institute, Jeonnam Agricultural Research & Extension Services, Iksan 58213, Republic of Korea; 3Ministry of Agriculture, Food and Rural Affairs, 94 Dasom 2-ro, Government Complex-Sejong, Sejong-si 30110, Republic of Korea

**Keywords:** rice Bakanae disease, drone, smart farm, paddy field agriculture, artificial intelligence, deep learning

## Abstract

This paper proposes a system for the forecasting and automated inspection of rice Bakanae disease (RBD) infection rates via drone imagery. The proposed system synthesizes camera calibrations and area calculations in the optimal data domain to detect infected bunches and classify infected rice culm numbers. Optimal heights and angles for identification were examined via linear discriminant analysis and gradient magnitude by targeting the morphological features of RBD in drone imagery. Camera calibration and area calculation enabled distortion correction and simultaneous calculation of image area using a perspective transform matrix. For infection detection, a two-step configuration was used to recognize the infected culms through deep learning classifiers. The YOLOv3 and RestNETV2 101 models were used for detection of infected bunches and classification of the infected culm numbers, respectively. Accordingly, 3 m drone height and 0° angle to the ground were found to be optimal, yielding an infected bunches detection rate with a mean average precision of 90.49. The classification of number of infected culms in the infected bunch matched with an 80.36% accuracy. The RBD detection system that we propose can be used to minimize confusion and inefficiency during rice field inspection.

## 1. Introduction

Rice is an essential food resource in Asia, and global concerns over rice self-sufficiency have increased as the topic of food sovereignty has been emphasized due to instabilities in trade associated with COVID 19 and the Ukraine–Russia war [1]. In addition to its role as a major resource for stabilizing food supply and demand, rice occupies an important position in terms arable land utilization and employment as well.

With recent dietary changes due to globalization, an increase in the consumption of high-quality rice has been documented [2,3]. Accordingly, seed quality control is increasing in importance. To this end, rice seed and field inspections are conducted in Korea. For the field inspections, cultivar purity, weed germination, and disease occurrence tests are performed on paddy fields that have been sufficiently isolated to prevent natural hybridization between different seeds. Since it is essentially an inspection of seed dissemination, field inspections are largely carried out during seeding seasons. However, the frequency and timing of inspection can be adjusted for early detection and removal of specific diseases [4]. There are various diseases that can occur in rice, namely blast disease, sheath blight disease, bacterial blight disease, rice stripe virus, and rice Bakanae disease (RBD) [5].

RBD is caused by a fungal pathogen. The disease continues to spread from the infected rice to the surrounding area. As the infected seedlings die or produce chaff, rice production is directly affected, and the fungus attaches to the normal rice seeds, causing the disease to reoccur when sowing in the following year [5]. Therefore, prevention of repeated infection cycles through early detection of RBD is essential for its effective control. Specifically, RBD-infected rice bunches are easier to distinguish from normal rice bunches before seeds are formed (Figure 1). Therefore, field inspections are conducted at that period. RBD field inspections generally calculate the morbidity rate based on the number of infected culms compared to the total rice field area, as determined by experts [6]. When inspecting an entire rice paddy field, inspection objectivity is often inconsistent due to expert bias, and is exacerbated by differences in weather, sunlight, location, or imaging angle [7]. Since multiple cross-validation by numerous experts is time-intensive and costly, automated detection of RBD is required to continuously achieve consistent test results and reduce resource costs [5,8].

Preprocessing for disease classification and detection is conventionally performed using plant imagery. Morphological information, such as the length of a plant or the width of a leaf, can also be directly collected by an observer. Image preprocessing into an advantageous form to achieve accurate identification is generally performed to enhance the differences between diseased and normal regions. Accordingly, classical preprocessing methods, such as support vector machine (SVM) or machine vision models have been employed [9,10]. However, since the distribution of preprocessed data is distinct from that of existing data, it may result in the over-fitting of more slight environmental changes [11].

RBD detection studies have commonly employed quantifying characteristics such as seedling height and width, the angle of the forked culm, Saturation from RBD images taken according to a certain standard, and SVM classification using the quantified data [5,12,13]. Importantly, it remains difficult to collect data in hard-to-access places, such as the interior of a rice paddy field. To this end, accessibility restrictions have been overcome by performing data collection via drones [8]. Accordingly, this prior study exploited the distinct color and Saturation corresponding to disease-affected parts of rice plants. Therefore, preprocessing was performed by applying a threshold value to the Hue and Intensity values of the image for increasing its contrast, followed by segmentation. Elsewhere, a deep learning object detection model was trained to detect the location of various plant disease parts such as apple black rot or tomato leaf spot within an image, and the resulting validity of the detection results was verified based on the learning results and visualization to produce a heat map of infection [14,15,16]. A further study on sequential systems performed disease detection over a wide range to select suspected infected areas, and detecting and classifying the diseased areas within these ranges [17,18]. The results showed that false detection over a wide range could be reduced, and the infection site could be more accurately detected, which inspired the two-step method of our study.

The present study aimed to devise a system for identifying the RBD-infected culms during field inspections for early detection and control. In order to analyze the infection rate via field inspections, precise areal calculations and detailed detections of culm numbers are required. Since the camera of the drone used for field inspection is a mono-vision camera, the area cannot be calculated alone. Therefore, to determine the exact area, image height and angle were specified through camera calibration, and data collection was limited to a controlled environment. For RBD infection detection, a two-step configuration was used to recognize the infected culms through deep learning classifiers. The YOLOv3 and RestNETV2 101 models were used to detect infected bunches and determine the infected culm numbers, respectively. In the process of classifying the number of infected culms in a bunch, focal loss function and transfer learning were adopted to resolve the data imbalance between each class.

## 2. Proposed System

For RBD-infected rice, aging progresses faster than normal. As a result, the infected rice grows thinner and longer than normal rice, while its color is yellower and less saturated than that of normal rice leaves, as shown in Figure 2. Assuming that such patterns are found in paddy fields, observations from above are advantageous, as the infected portions are covered by normal rice when viewed from the side.

In general, deep learning models exhibit higher levels of performance with an increase in training data amount. Therefore, certain limitations can be overcome by increasing the training data amount. Since the collection and labeling of drone images is both time and labor intensive, it remains a challenge to collect sufficient data for adequate generalization of the deep learning model across all height and angle imaging conditions. Furthermore, since the automation of rice field inspection to be performed encompasses the total rice-planted area, maintaining the imaging height and angle constant is advantageous. When detecting RBD from images collected via drone, symptoms may or may not appear clearly, depending on the specific imaging height or angle, in conjunction with the environmental conditions. Therefore, in the present study, the optimal shooting height and angle for RBD detection were selected. Subsequently, through camera calibration and the construction of a viewpoint transformation matrix, it was possible to calculate a specific area at the selected height and angle [19,20].

Rice bunches were positioned at equal intervals (Figure 3). A single bunch of rice contains several stems that can bear seeds (Figure 4), which are defined as individual “culms”. During field inspection of the RBD-infected culms, calculations were performed by summing the number of infected culms, whereas when the detection model was trained through the RBD data collected at the selected shooting height and angle, the output value refers to the infected bunch, so additional analyses are required. Therefore, in this study, the RBD detection model was configured in two stages: to find the candidate regions for RBD infected bunches via the deep learning model, and to classify the number of culms infected with the disease.

Figure 5 describes the system proposed for the analysis of the RBD-infected rate based on the aforementioned related research and prior information. The proposed system configuration preceded RBD image collection using drones. The data for deciding optimal shooting height and angle were collected for each condition across specific vertical heights and angles. Subsequently, the optimal vertical height and angle values were inferred by analyzing the data for each condition. Afterward, camera calibration was performed to minimize the effects of distortion, and automatically calculate the ground area under inferred vertical height and angle. While collecting data inferring the number of infected culms, RBD-containing images were collected under optimal shooting conditions, and bounding boxes of bunches were labeled and the number of culms counted, using the images. The labeled data were used to train a detection model for extracting the total area of RBD bunches, and to train a classification model to identify the number of infected culms contained within the infected bunches. When the configuration and learning of each module was completed, the entire system detected the RBD bunch area with respect to the area of interest from the drone image. By performing classification based on the detected bunch areas, it is possible to quantitatively calculate the number of RBD-infected culms relative to the total area.

### 2.1. Selection of Drone Shooting Height and Angle Using Linear Discriminant Analysis and Gradient Intensity

If the cost of data collection and labeling is low, the data collected from all shooting heights and angles can be used as input data for training RBD detection and classification models. However, since deep learning models generally require a lot of data, it is practically impossible to collect data in all cases. Therefore, the optimal shooting height and angle were inferred using a simpler model in the present study, focusing on the morphological aspects of RBD (Figure 6). The collected data were cropped. Specifically, a single bunch of RBD infections observed at the same location in 16 separate imaging conditions was cropped to fit. Additionally, a corresponding normal rice bunch that was the same size as the infected bunch was cropped, which yielded 16 RBD bunches and 16 samples of normal rice bunches corresponding to each height and angle. In total, 91 sets of data were collected, consisting of a total of 2912 (16 × 91 × 2) image data points. Figure 7 displays a set of example images in which the area infected with RBD, and normal rice areas were cropped out.

The cropped images of the RBD and normal rice bunch areas have different resolutions, respectively. However, to analyze this with linear discriminant analysis (LDA) and gradient magnitude, all images must have the same resolution. Therefore, all images were resized to 29(W) × 29(H), which was the smallest resolution among all images. LDA and gradient magnitude were used to quantify RBD. Specifically, LDA was used to compare the color information of RBD. LDA is a linear dimension reduction supervised learning method. Converting an image from the RGB color space to the HSI color space gives requires the calculation of Hue and Saturation (Equations (1) and (2)):(1)$$Hue=\cos^{-1}\frac{0.5\times \left(\left(R-G\right)+\left(R-B\right)\right)}{\sqrt{{\left(R-G\right)}^{2}+\left(R-B\right)\left(G-B\right)}}$$
(2)$$Saturation=1-\frac{3}{R+G+B}\min\left(R,G,B\right)$$

As stated, it is possible to use the distinct yellowish color and lower Saturation compared to normal rice to distinguish RBD. Accordingly, Hue can be advantageous in the extraction of RBD information, whereas Saturation can help identify areas of low Saturation. Of the 2912 normal and RBD bunches image data, each image became a 1682 (29 × 29 × 2) dimensional vector, which was divided into two classes—RBD and normal—via LDA to create a one-dimensional projection space. Scalar values can be obtained by projecting the data for each of the 16 shooting environments into a one-dimensional projection space. Scalar values computed from 2912 image data points belong to one of 16 classes. The calculated scalar values were normalized using the Mahalanobis distance or Batarachaya distance, and then the average value was used to obtain the recognition degree (Figure 8a).

Gradient magnitude was used to calculate the average texture feature values when images contained RBD, as it was expected that edges with a stronger Intensity than that of normal rice would be included in terms of edge detection. Accordingly, the Intensity was obtained using Equation (3):(3)$$Intensity=\frac{R+G+B}{3}$$

The gradient values in horizontal and vertical directions are obtained by applying an edge filter to the Intensity. Accordingly, the magnitude values can be acquired using the Euclidean distance from the gradient values in the two directions (Figure 8). The gradient magnitude value is the average value of the gradient magnitudes of each pixel in the 29 × 29 image, and it corresponds to the right portion of Figure 8. The gradient magnitude values of the RBD and normal rice classes were 21.94 and 20.67, respectively, indicating that the RBD class had a high gradient magnitude value.

Since the one-dimensional LDA and gradient magnitude values obtained using Hue, Saturation, and Intensity maintain different scales, the values were normalized by converting them to a standard normal distribution, with a mean of 0 and a variance of 1 using the whitening formula shown in Equation (4):(4)$$\tilde{x}=\frac{x-\mu }{\sigma }$$
where x is the data before whitening, μ is the average of the data, σ is the standard deviation of the data, and $\tilde{x}$ is the whitened data. The average value of each whitening-processed one-dimensional LDA and gradient magnitude value became the final recognition degree.

### 2.2. Camera Calibration

Due to resolution limitations, the drone-attached camera could not accurately calculate the number of rice plants planted in the paddy fields, no matter which algorithm was applied. Therefore, it was impossible to calculate the infected rate of RBD by bunches or by culms based on imagery information alone. Due to the mechanization of agriculture, however, the distances between rows and columns of transplanted rice were standardized to 20 cm and 30 cm, respectively (Figure 3), and accordingly, it became possible to calculate the amount of rice grown per area. Specifically, 16.67 bunches were cultivated per 1 m^2^, and an average of 23 rice culms were grown per bunch, resulting in 383.33 culms cultivated per 1 m^2^. Based on this information, accurate camera calibrations could calculate the RBD-infected.

### 2.3. RBD Bunch Area Detection and Culm Classification

Characteristic symptoms of include plants growing 1.5 times taller than normal during the growing season, as well as the death of the culm as the leaves turn light green [21]. Detecting RBD from drone imagery based on these features is across various environments is thus complicated due to its relative similarity to normal, non-infected rice. Furthermore, the inconspicuous nature and relatively low areas of RBD occurrence also inhibit the efficient extraction of information. Accordingly, to reduce the misdetection of RBD, it was necessary to sacrifice the processing speed instead of increasing the accuracy of the algorithm. In addition, considering that the field inspection of RBD-infected was carried out in the middle of a rice field, it was difficult to operate equipment requiring additional electricity supply. However, it remains necessary to present the results onsite wherever possible for efficient exchange of information. Therefore, a deep learning object detection model with high accuracy and fast detection speed, even across various and complex environments, was selected. In addition, to identify the number of RBD-infected culms across the total area of field inspection, classifications of culms with RBD-detected bunches were run as well. Here, a series of interwoven processes for the proposed RBD detection are shown in Figure 9, as well as the number of RBD culms in the detected bunch region.

#### 2.3.1. Model Structure

First, five candidate deep learning object detection models were selected for pre-evaluation of RBD-infected bunch region detection. Based on a comparison of mean average precision (mAP) and frames per second (FPS) in the open Pascal VOC 2007 dataset [22], the YOLOv3 model was selected.

Table 1 shows the performance comparison information of the deep learning object detection models for the dataset.

The YOLOv3 model was configured to detect the area bunches of RBD (Figure 9). Since a large amount of high-quality data is required when training deep learning object detection models, transfer learning [30,31,32] was performed using the weights of the DarkNet 53 model pre-trained with ImageNet [33], an open dataset used for image recognition. Since the pre-trained DarkNet 53 model used an input with an aspect ratio of 1:1, a cropped image of size 3208×555 was divided into size 555×555. To minimize the loss occurring in the boundary region when dividing the image, a patch image was generated by overlapping a certain region in the form of a sliding window (Figure 10).

#### 2.3.2. Classification of Culm Number in the Detected RBD-Infected Bunch Area

For the bunch areas detected through the YOLOv3 model, the images of the bunch areas were extracted from the bounding box coordinates predicted from the input image, and then the number of culms was classified by going through an additional convolutional neural network (Figure 9). When using this method for RBD identification, the ambiguity of culm classification, as well as locations where RBD bunch regions do not show clear features due to changes in weather or light sources, can affect the classification performance. As shown in Figure 7a, an RBD-infected culm is composed of one or more leaves. It should yield only one infested culm, even if two or more leaves appear to be infected. However, inaccurate labeling can lead to a significant drop in classification performance with a single infected culm misclassified as two or more.

To prevent performance degradation due to such ambiguity, consistent expert labeling is required. Here, the width and height of the RBD bounding box detected was not uniform. Accordingly, bounding boxes of various sizes were resized so that the width and height were 1:1, and they could then be input into the classifier. In the present study, a classification experiment was conducted using the deep residual network model with an input size of 224×224 pixels. Deep residual networks can be used to resolve deep learning model problems, such as vanishing gradients, exploding gradients, as well as increasing parameter numbers for increasing the number of layers of a neural network using a residual block structure, and 1×1 convolution structures [34]. With such characteristics, the model showed high performance feature extraction and classification. Accordingly, since the deep residual network has the potential to show high performance while readily controlling the speed and performance according to the increase in parameter number and layers by changing the number of repeated structures, this model was used for culm number classification.

## 3. Experiments and Results

### 3.1. Experiment Environment

In this paper, the DJI ‘Mavic2 Pro’ drones were operated to collect data for the detection of RBD under different environmental conditions, such as different illumination, diverse seasons, or weather conditions. The camera installed in the device has a resolution of 5472 pixels horizontally, 3648 pixels vertically, and a field of view of 77°. When the shooting view was calculated according to the technique shown in Figure 11, where the drone is operated with a 3 m vertical height, and a downward shooting angle of 0°, an image that detects 9 m left and right at a minimum detection distance of 6.8 m can be obtained.

Experiments and evaluations were performed on a workstation in Windows 10 environment, equipped with Intel Xeon(R) Gold 6230 R CPU 2.10 GHz and 128 GB RAM. Each module for the experiment was implemented in Python 3.6. Tensorflow 1.15 library was used for training and evaluation of deep learning algorithms, and 2 GeForce RTX 2080TI 11 GB GPUs were used for learning and evaluating each deep learning module.

### 3.2. Dataset Details

#### 3.2.1. Drone Shooting Height and Angle Selection

Table 2 shows the composition of the dataset collected for selecting the imaging height and angle.

In order to increase accuracy, various environmental conditions were included in the data. For this purpose, images were collected from July to August 2020 in Namwon-si, Gimje-si, and Gunsan-si, Jeollabuk-do, Republic of Korea, with all areas corresponding to ordinary paddy fields in the plain area. Drone imaging vertical heights from the ground were set at 1, 2, 3, and 4 m. When the observation height exceeded 4 m, RBD was not visually identified due to the drone camera resolution. Therefore, only heights up to 4 m were used in the experiment. Furthermore, if the shooting down angle exceeded 15°, diffuse reflection would be induced, making it challenging to distinguish between RBD-infected and normal culm using the naked eye. Therefore, shooting angles were set to 0, 5, 10, and 15°.

The imaging was performed on various days so that various weather conditions were included under a range of back light and front light conditions. To observe the disease of rice in the same bunch, the drone remained focused in the same position and direction to take pictures in the order of 0°, 5°, 10°, and 15° at 1 m, and in the same order at 2, 3, and 4 m (Figure 12). When shooting, the shutter was pressed twice at a specific height and angle to take a picture, to avoid instances where infected plants cannot be identified because rice is bent by the wind, and shooting failure due to user error or data transmission error between drone and remote control. The normal number of shots was 126. However, for the reasons mentioned above, 124 to 128 images were acquired per condition, as shown in Table 2.

In the experiment, the same location RBD-infected bunch sample was labeled across all conditions. Since there few samples existed in the same location, 91 matched data were obtained for each height and angle condition. Similarly, normal bunches with the same location were also collected around 91 infected bunches under 16 conditions. Accordingly, 1456 normal bunches of data for 16 conditions, and 1456 RBD-infected bunch data were used in the analysis.

#### 3.2.2. Detecting RBD Bunch Areas and Culm Classification

Data collection was performed from July to August 2021 for training the deep learning model for detecting RBD-infected bunch area and culm classification. Here, images collected from Jeollabuk-do, Gyeongsang-do, and Chungcheong-do Republic of Korea were used. All areas correspond to ordinary paddy fields in the plain area. The total number of images collected for training the deep learning model is shown in Table 3.

The collected 20,514 images contained data making it difficult to identify RBD, such as images taken under strong winds, or blurred focus due to photographer error. In addition, data in which the identification of RBD bunches was ambiguous or infeasible due to various factors, or data that did not include actual RBD bunches were included. As a result of removal such unrecognized images, 10,003 usable images were obtained.

Random cropping was performed from the 10,003 datasets (Figure 13). This cropped image was labeled by both experts and non-experts, and the bounding box labeling was performed via expert inspection for the amount of work of non-experts. Thereafter, dividing the cropped data of 10,003 sheets through a sliding window (Figure 10), 28,365 pieces of RBD-containing cropped patch image data were generated. Cropped patch image data were then used for training and evaluation of the detection model, where 26,812, 769, and 784 pieces of data were used for model training, validation, and evaluation, respectively.

In order to obtain the culm number classification model for RBD, it was necessary to also classify the number of culms in each bounding box (Figure 14). As mentioned in Section 2.3.2, this required expertise, and thus is an intensive task for an expert to process all data. Therefore, labeling was performed by non-agricultural experts, and then reviewed by experts. Table 4 shows the total amount of data collected using this method. In July and August, when disease identification is easier, it is common that the disease does not spread to the entire bunches. Therefore, it can be seen that the data were biased in the range of 1 to 5 culms of infection.

### 3.3. Quantitative Evaluation Indicators

The evaluations were based on two metrics: mAP and a confusion matrix. mAP is an index used to evaluate the performance of a deep learning object detection model, and is based on average precision. Here, only the evaluation based on IoU 0.5 was used for assessing PASCAL VOC (Equation (5)), where the average precision corresponds to the area under the precision-recall graph, while the recall maintains a regular interval, such as [0,0.1,0.2,…,1]. In Equations (6) and (7), $P_{interp}\left(r\right)$ is the maximum precision for recalls > r, $P\left(\tilde{r}\right)$ is the precision evaluated at recall $\tilde{r}$, and AP is calculated as the average of $P_{interp}\left(r\right)$ over all recall ranges:(5)$$IoU=\frac{area\left(gt_{box}\cap^{\;}tr_{box}\right)}{area\left(gt_{box}\cup^{\;}tr_{box}\right)}\{\begin{array}{c} IoU\geq Threshold\left(0.5\right):Pos\\ IoU<Threshold\left(0.5\right):Neg \end{array}$$
(6)$$P_{interp}\left(r\right)=\max_{\hat{r}\geq r}P\left(\tilde{r}\right)$$
(7)$$AP=\frac{1}{11}\underset{r\in \left\{0,0.1,\ldots ,1\right\}}{\sum }P_{interp}\left(r\right)$$

### 3.4. Drone Shooting Height and Angle Selection

Figure 15 shows the recognition degree obtained by calculating sample data for a total of 16 shooting conditions according to the algorithm shown in Figure 6. It can be seen that the value of one-dimensional LDA reached its maximum of 1.255 at 3 m and 10°, whereas the value of gradient magnitude peaked (1.312) at 3 m and 15°. Lastly, for the one-dimensional LDA and the average of the gradient magnitude values, the recognition degree of RBD reached its maximum (1.23) at 3 m and 0°. Figure 16 shows the results of visualizing the images by projecting the vectors for the image onto a one-dimensional straight line via LDA. It is readily apparent that few data exist that are likely to be confused at 3 m and 0° values near the class boundary.

### 3.5. Camera Calibration

A large calibration banner measuring 8 × 6 m (width × height) was produced considering that 9 m left and right was imaged at a minimum sensing distance of 6.8 m (Figure 11). To identify the optimal detection position, the drone was fixed on a ladder, and images were taken at 1 m intervals from 6 to 20 m (Figure 17). As shown in Figure 11, the front part of the calibration banner is missing from the field of view at 6 m, whereas the banner is completely within the camera field of view at ≥7 m.

Figure 18A shows the image of the calibration banner at a distance of 7 m from the camera attached to the drone. The red gridded lines and dots in Figure 18A indicate intersections in the calibration banner, and the red line extending vertically is a straight line that crosses the center of the calibration banner in the longitudinal direction, whereas the blue vertical line crosses the center of the image in the longitudinal direction, revealing that the calibration banner was not placed exactly in the center of the image. To compensate for this, perspective transformation was used. In Figure 18A, with horizontal and vertical dimensions of 5472 and 3648, respectively, the upper-left coordinate of the intersection point of the calibration banner is located at (1329, 2874), while those of the upper-right, lower-left, and lower-right coordinates are (3489, 2872), (759, 3416), and (3990, 3386), respectively.

Perspective transformation shifted the ground to the coordinate plane, the horizontal and vertical sizes to 20,000 and 22,000, respectively, the camera coordinates to (7000, 14,000), and the upper-left, upper-right, lower-left, and lower-right coordinates to (4000, 2000), (10,000, 2000), (4000, 6000), and (10,000, 6000) respectively (Figure 18B).

As mentioned, since the center of the calibration banner does not coincide with the center of the image, the calibration area was slightly biased to the left (Figure 18B). To compensate for this, the distorted angle was calculated using the resulting angle from the straight line between the center of the camera and the calibration banner, as well as the camera central axis in the perspective-transformed world coordinate system. The resulting distortion angle calculated in the analysis was 3.77°.

A region calibrated with respect to the central axis of the camera was obtained via rotation transforming the calibration area by a distorted angle with respect to the central axis of the camera, and converting it into a pixel coordinate system using inverse perspective transformation. Figure 18C illustrates the result of correcting the calibration area using rotation transformation in the world coordinate system.

Figure 18D shows the results of image correction based on the information of the calibration banner, and the camera central axis included in the perspective transformation of the pixel coordinate system data belonging to the drone image in the world coordinate system. The resulting coordinates of the calibration areas were upper left (1660, 2858), upper right (3808, 2891), lower left (1127, 3377), and lower right (4338, 3425), while the green rectangle in Figure 18D contains these areas, and can be used as the input for object detection. Here, the blue rectangular areas depict that of one piece of checkerboard used for calibration (equivalent to 1 m^2^). Therefore, it was possible to precisely calculate the area for 24 m^2^.

### 3.6. Detection and Classification of RBD

In order to verify the performance of the proposed system, the details of performance evaluation for each module of RBD bunch area and infected culm number detection were assessed. Culm classification using deep learning model structure included in the system consisted of two steps (Figure 9). Since end-to-end learning could not be performed because it included the step of projecting and extracting information of the detected region to the input image (as described above), different datasets were used for training and evaluating each module.

#### 3.6.1. RBD Bunch Area Detection

The training and evaluation data for the YOLOv3 model for RBD bunch area detection contained a total of 28,365 images, consisting of 26,812 training images, 769 verification images, and 784 evaluation images. During analysis, an augmentation method was applied to avoid over-fitting when training the YOLOv3 model. The weighting and batch size of the DarkNet53 network pre-trained with 9 anchors and ImageNet datasets determined in the experiment were set to 5.

Here, when analyzing the dataset for detecting the area of RBD bunches, there was an imbalance [35,36] between the foreground and background regarding the number of RBD-infected culms and normal rice. To address this problem, a focal loss function [10,37,38] was introduced, and parameter values of α=0.25 and γ=2 were applied. In addition, a gradual learning rate reduction was applied using a cosine learning rate decay method [39]. The training parameters of the YOLOv3 model for detecting RBD bunch areas were set as shown in Table 5 by adjusting the batch size, epochs, learning rate, etc., and adding cosine attenuation, in accordance with previous research related to apple detection [40].

The experiments were conducted on the application warm-up techniques and label smoothing [41,42] during the training process using the learning parameters presented in Table 5. For the warmup, issues were addressed regarding the problem of excessive gradient values occurring at the beginning of learning due to the data being biased toward normal data, as well as the issue of excessively increasing confidence during the detection of RBD ambiguous features using label smoothing. As label smoothing was applied, the data were augmented [43,44], and the deleterious effects of data imbalance problem can partially alleviated. The performance comparison evaluation was conducted using the mAP value based on an IoU threshold value of 0.5. As a result of applying warm up and label smoothing in the corresponding experiment, it was confirmed that the performance increased by 9.16% compared to when they were not applied (Table 6). Accordingly, it was presumed that the regularization effect of the data was achieved by suppressing the over-fitting of inaccurate training data.

Figure 19 shows the comparison results of the same YOLOv3 model with and without the warmup and label smoothing techniques applied. Specifically, Figure 19A shows that when detecting RBD bunch area through the learning results without applying warm up and label smoothing techniques, missed detection was more frequent. In contrast, Figure 19B shows relatively more fitting detection results and avoidance of missed detections, as a result of training using warm-up and label-smoothing techniques.

Figure 20 presents the non-maximum suppression result image with a threshold value of 0.45 for the bounding box, pertaining to the bunch area of RBD detected through the YOLOv3 model for cropped patch data. The yellow bounding box indicated the actual RBD bunch areas, whereas the red bounding box indicates the predicted results of the RBD bunch areas. In spite of the trainings applying the data reinforcement mix-up method, the warm-up method, and the label smoothing method, misdetection could be confirmed.

#### 3.6.2. Classification of Culm Number in the Detected RBD Bunch Area

Based on the YOLOv3 model producing 90.49% detection performance for the evaluation data (Figure 9), the RBD bunch areas were detected, and extracted through the bounding box coordinates. After reconstructing the RBD bunch area to a size of 224×224, infected culm number classification was performed through the deep residual network. The number of culm classification learning and evaluation data in the RBD bunch area total 31,826 images. To train and evaluate the deep residual network, labeling was performed according to the number of RBD culms, and consisted of 78% training, 20% validation, and 2% testing. Table 7 shows the dataset for deep residual network training and evaluation.

The classification performance of RBD was evaluated using the relationship between the actual and predicted classes according to the data in Table 4. Table 8 shows the deep residual network performance by network depth for the evaluation data. In general, as the depth of the convolutional neural network increased, the expression level of the filters used varied, providing high performance feature extraction. However, the result of the deep residual network with 50 layers was 80.36%, approximately 9.91% and 4.41% higher than that of the deep residual network with 101 and 152 layers, respectively (Table 8). Accordingly, performance verification was performed on the detection of RBD in the deep residual network with 50 trained layers.

The data in Table 9 confirms that the classification performance with ≥6 RBD-infected culms was ≤70%. Although the focal loss was applied to the difficulty of data augmentation due to the lack of data, a data imbalance between classes was confirmed. Figure 21 shows the results of predicting RBD-infected culm numbers using the learned classification results for 50 layers of the deep residual network.

In the previous experiment, 1–8 RBD-infected culms were included in the bunch area, and produced an accuracy of 80.36%. However, there is notable room for improvement regarding classes with limited data. To confirm this, a classification analysis was conducted using only all classes except 6–8 culms, which maintained the smallest number of classification data, and the results when including only 1–5 infected culms are shown in Table 10.

In order to perform a comparative experiment based on the depth of the deep residual network on the reconstructed RBD culm classification learning and evaluation data, the learning parameters were set and trained according to Table 5. Table 11 shows the performance comparison results for the RBD classification data according to the reconstructed network depth. The performance of the deep residual network with 101 layers was 85.36%, representing increases of approximately 3.64% and 0.92%, when compared with the deep residual networks with 50 and 152 layers, respectively. Therefore, performance verification was conducted for each number of RBD on the learned deep residual network with 101 layers.

The data in Table 12 confirm that the classification performance for all number of culms was ≤80%. Compared with the experiment in which the data were divided into 1-8-culm classes, it can be confirmed that the performance degradation caused by data imbalance between classes is improved. Figure 22 shows the predicted number of RBD-infected culms for 101 layers of the deep residual network of the reconstructed classification data.

Figure 23 compares the classification performance for the number of RBD culms with respect to Table 8 and Table 11. The performances following reorganization of the number of RBD culms from 1 to 5 were improved by 1.91%, 11.29%, and 13.91%. Therefore, it was concluded that it was derived by resolving the issue of imbalance in the amount of data.

## 4. Conclusions

This study proposed an automatic detection system using drone imagery and deep learning models for the early detection of RBD, and automation of rice field inspections. Prior to training the deep learning model, the LDA and gradient intensity values of the data were collected according to certain rules were analyzed, paying particular attention to the color and morphological characteristics of RBD. Subsequently, at the selected optimal 3-m height and 0° angle, a viewpoint transformation matrix was derived for the correction of camera distortion and area calculation. The YOLOv3 deep learning model was selected as the module for detecting the RBD-infected bunch area, and the training result had a mAP of 90.49. RBD-infected culm number classification was performed using the ResNet 50 layer model, with an accuracy of 80.36% achieved when labeling data for culms 1 to 8 were used. As a result of the experiment by reducing the class range from 1 to 5, the class imbalance problem was somewhat resolved and the accuracy increased to 85.36 in the ResNet 101 layers model. Therefore, it is expected that maintaining only 5 classes will have a stable effect on classification until sufficient data on rice bunches with ≥6 culms are obtained.

## Figures and Tables

**Figure 1 sensors-23-00032-f001:**
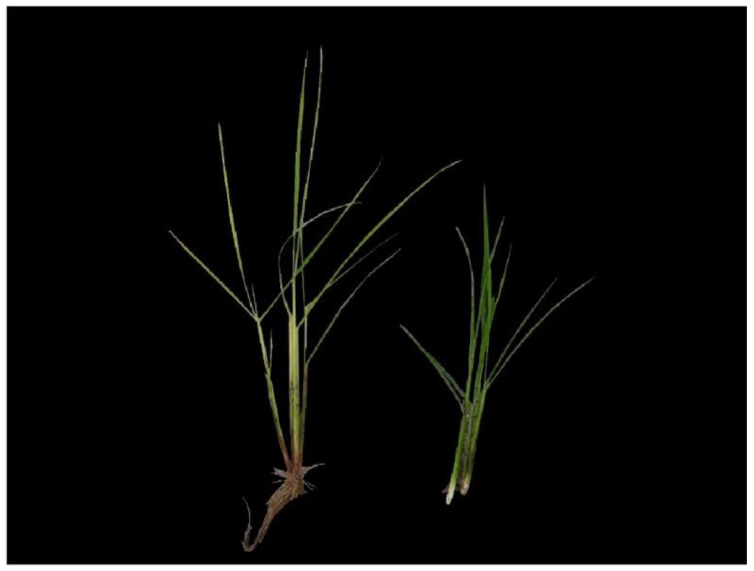
Rice infected with Bakanae disease (left) and normal rice (right).

**Figure 2 sensors-23-00032-f002:**
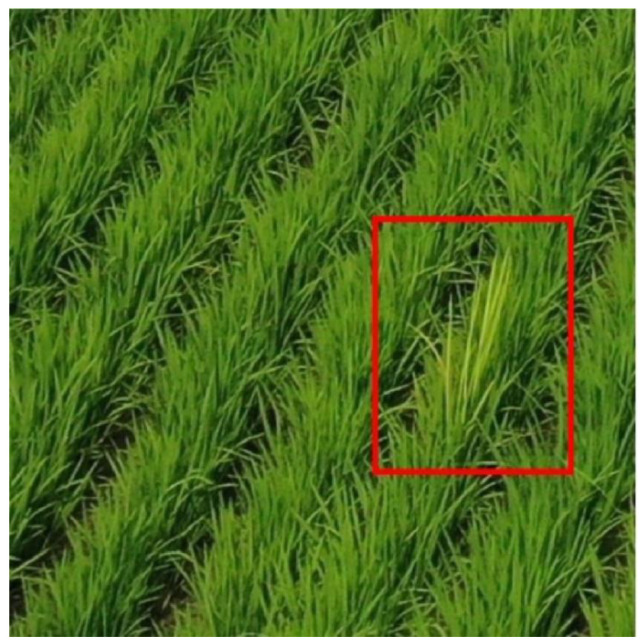
The rice infected with Bakanae disease photographed from above (Red box corresponds to RBD-infected bunch).

**Figure 3 sensors-23-00032-f003:**
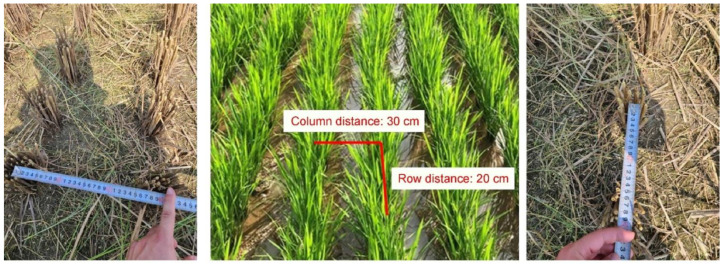
The standardized rice transfer interval due to agricultural mechanization.

**Figure 4 sensors-23-00032-f004:**
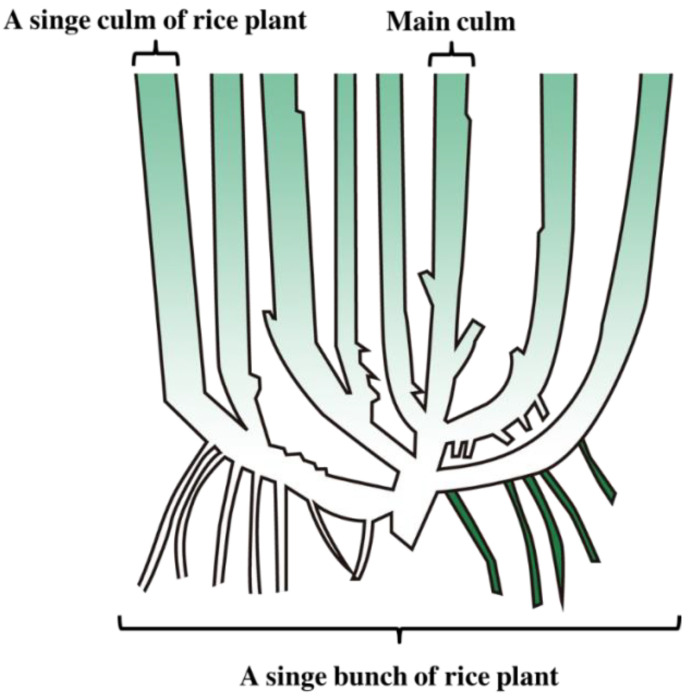
Example bunch and culm of rice plant.

**Figure 5 sensors-23-00032-f005:**
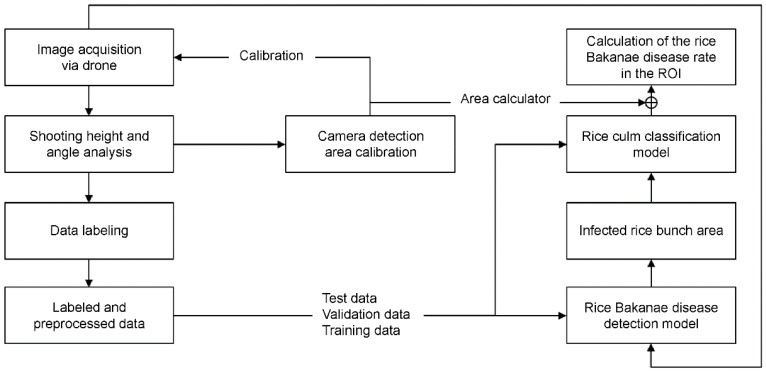
The system structure for reading the RBD-infected.

**Figure 6 sensors-23-00032-f006:**
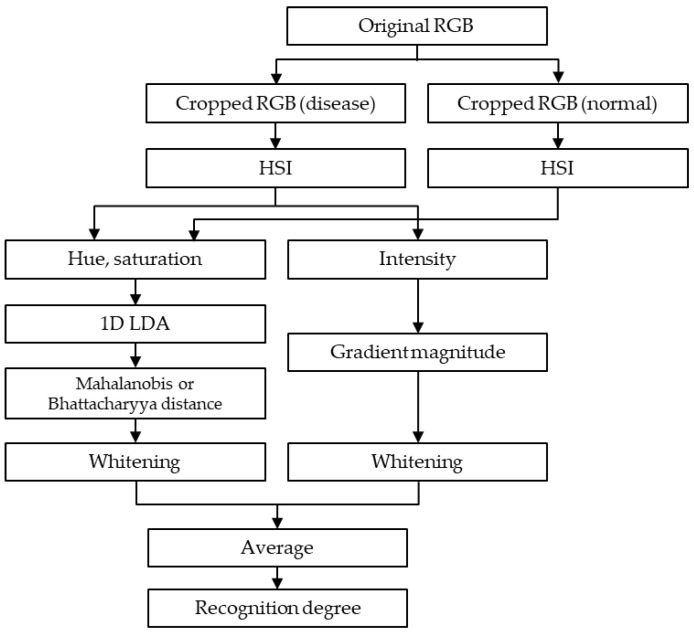
The algorithm flowchart for calculating RBD recognition degree.

**Figure 7 sensors-23-00032-f007:**
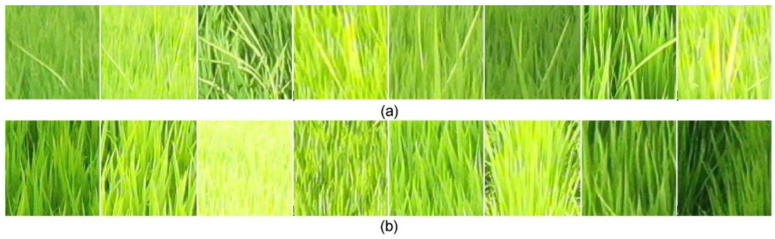
An example of image cropping RBD and normal rice plants: (**a**) Images in which the area containing rice infected with Bakanae disease is cropped, and (**b**) images in which an area containing normal rice is cropped.

**Figure 8 sensors-23-00032-f008:**
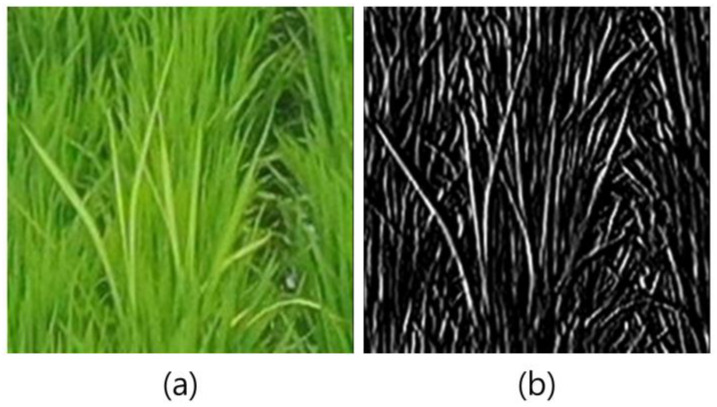
The edge detection of images containing RBD: (**a**) Original image, and (**b**) edge image containing gradient magnitude values.

**Figure 9 sensors-23-00032-f009:**
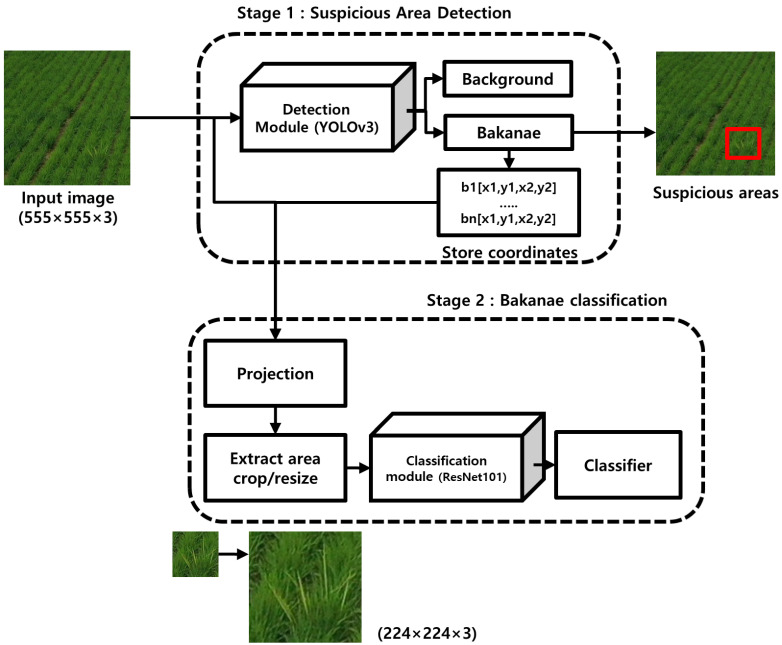
The process of determining the number of RBD-infected culms.

**Figure 10 sensors-23-00032-f010:**
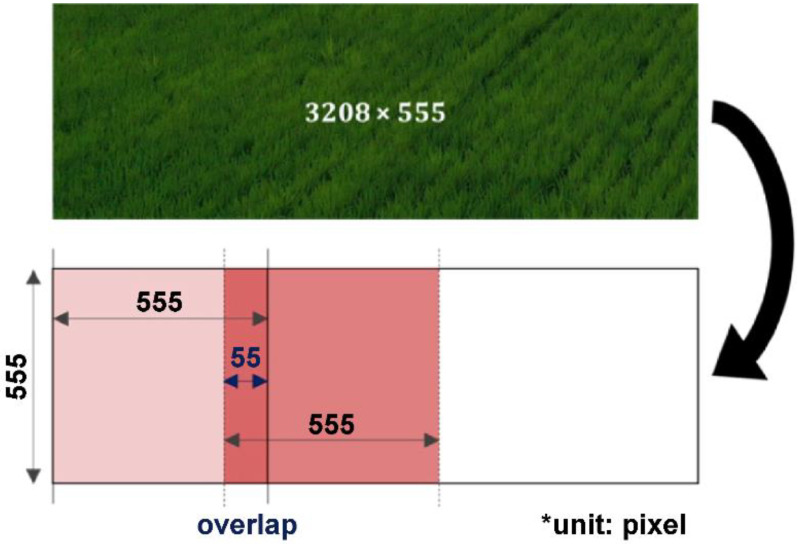
The method of creating patch image data using sliding window.

**Figure 11 sensors-23-00032-f011:**
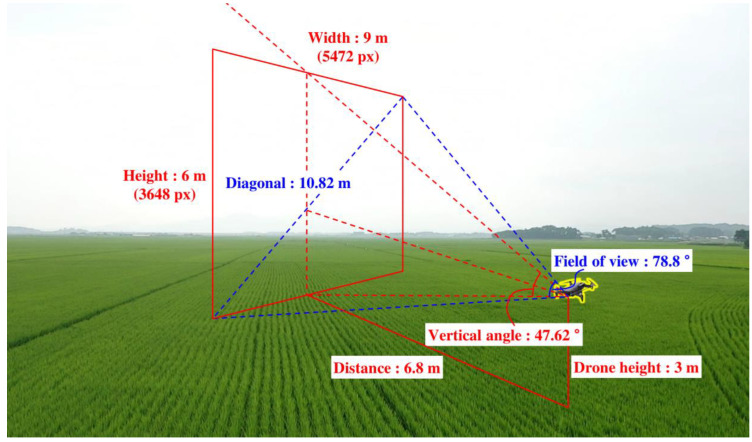
Calculation of the camera field of view for the operating drone.

**Figure 12 sensors-23-00032-f012:**
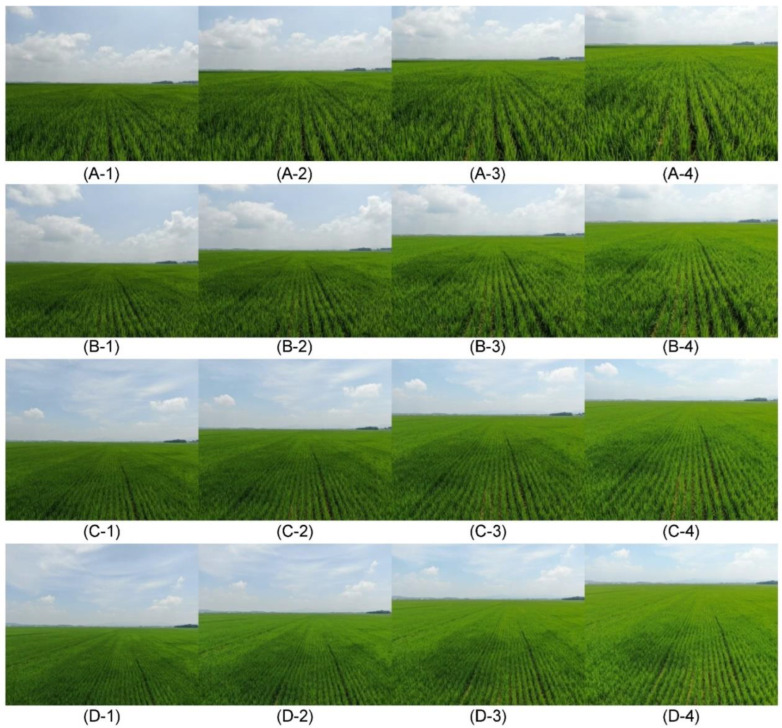
Drone images collected from various height and angles conditions (**A**–**D**) heights of 1, 2, 3, and 4 m, respectively; (**1**–**4**) 0°, 10°, 20°, and 30° down angles, respectively.

**Figure 13 sensors-23-00032-f013:**
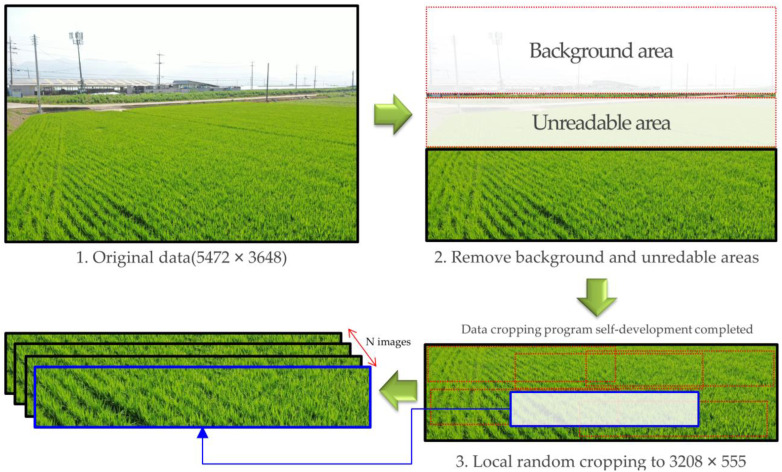
Random cropping procedure from original image.

**Figure 14 sensors-23-00032-f014:**
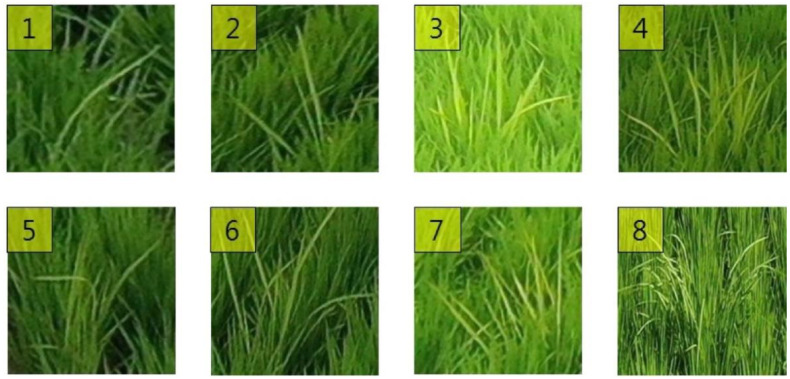
Example of data by number of culms of RBD (The numbers in the upper left mean the actual number of culm).

**Figure 15 sensors-23-00032-f015:**
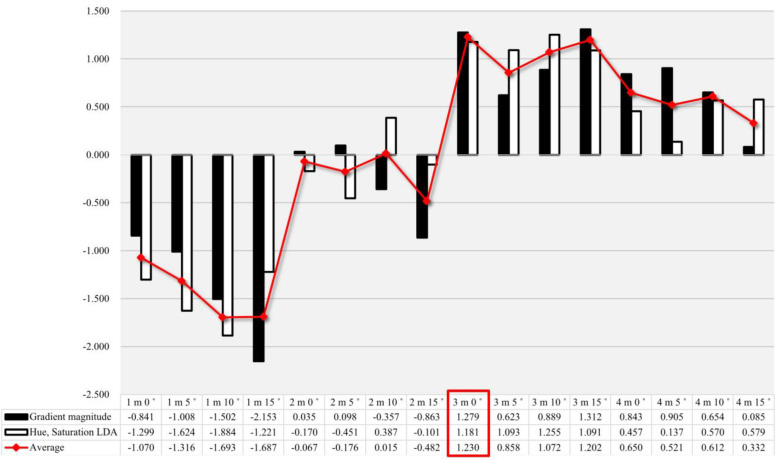
The rest results of recognition degree for RBD based on the selection of shooting height and angle (The red box indicates the height and angle at which the mean of the gradient magnitude, hue, and saturation LDA is the highest).

**Figure 16 sensors-23-00032-f016:**
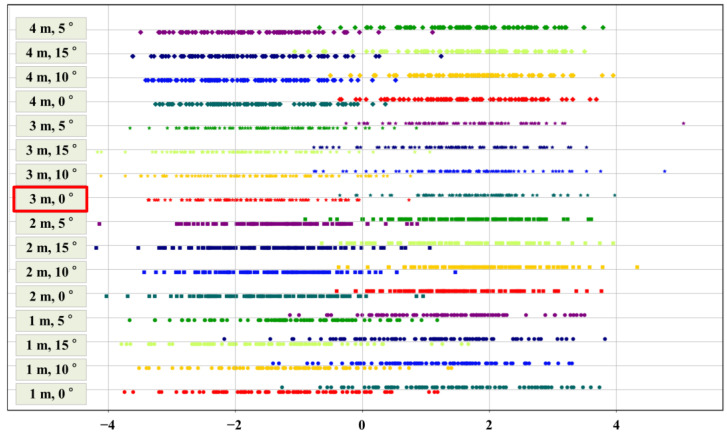
Visualization of LDA results by shooting conditions (The red box indicates the height and angle at which the mean of the gradient magnitude, hue, and saturation LDA is the highest).

**Figure 17 sensors-23-00032-f017:**
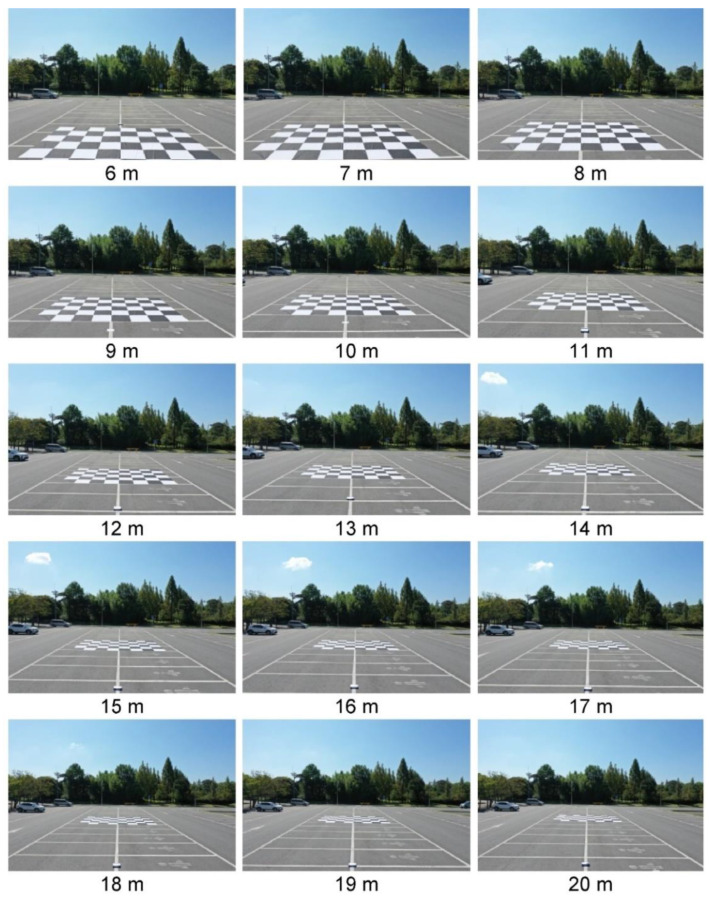
The images taken for camera calibration.

**Figure 18 sensors-23-00032-f018:**
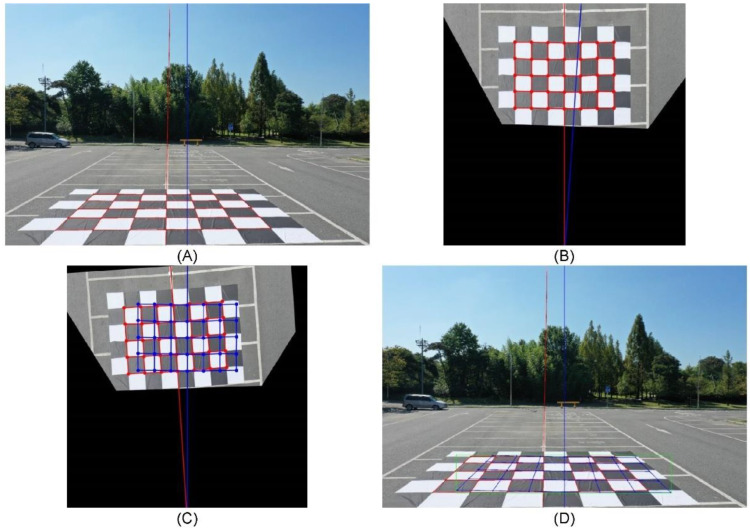
The perspective transformation for camera calibration: (**A**) reference lines and points for performing perspective transformations, (**B**) image projection from pixel coordinate to world coordinate, (**C**) center compensation based on the camera center point, (**D**) calculated recognition range using camera calibration.

**Figure 19 sensors-23-00032-f019:**
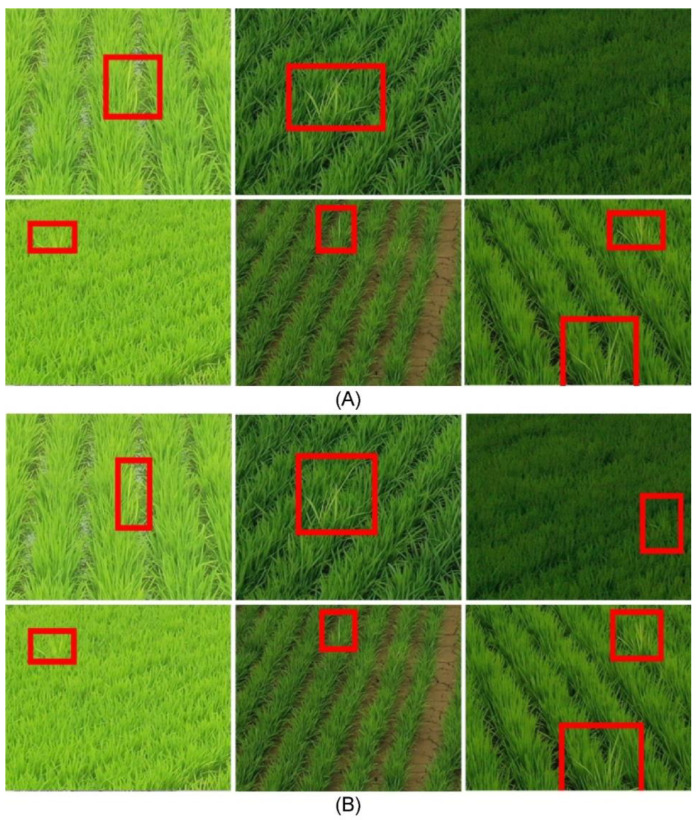
The detection results of RBD bunch area for patch segmentation data (Red boxes represent detected RBD-infected bunches): (**A**) warm up and label smoothing technique not applied, and (**B**) warmup and label smoothing techniques applied.

**Figure 20 sensors-23-00032-f020:**
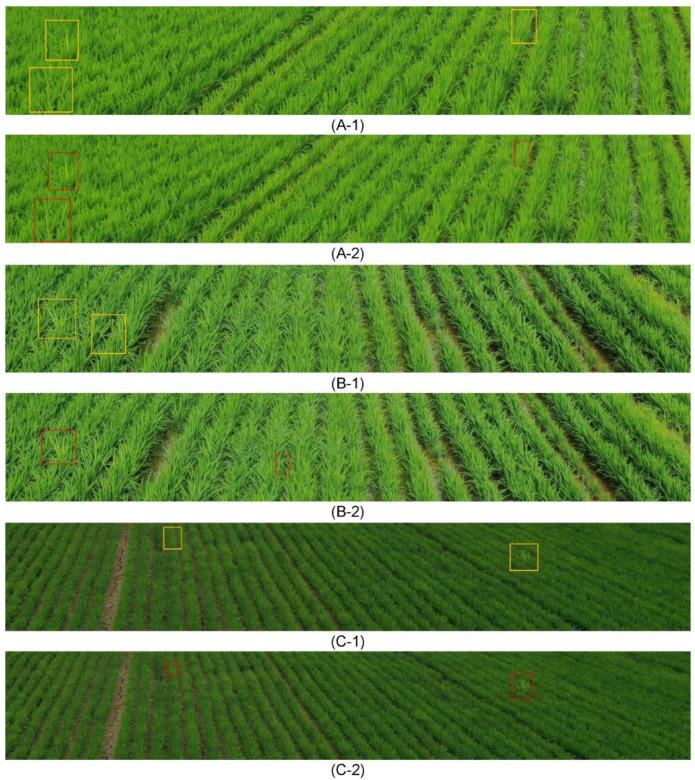
YOLOv3 model-based RBD bunch area detection results (Yellow boxes represent the ground truths, and red boxes represent the detection results): (**A**–**C**) Each detection task, (**1**) Ground truth, and (**2**) predicted result.

**Figure 21 sensors-23-00032-f021:**
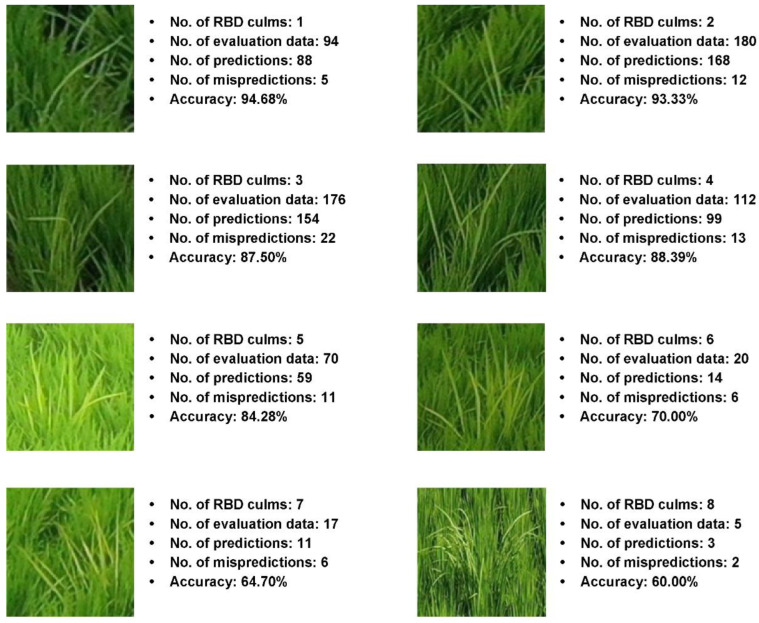
The prediction result of RBD for 50 layers of deep residual network.

**Figure 22 sensors-23-00032-f022:**
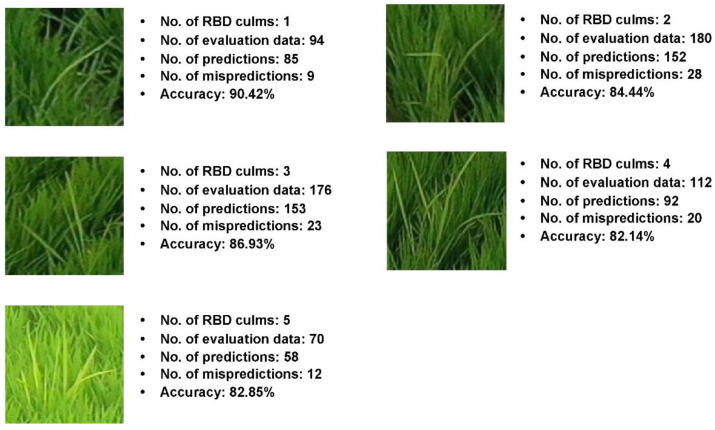
The prediction result of RBD based on reconstructed number of culms data.

**Figure 23 sensors-23-00032-f023:**
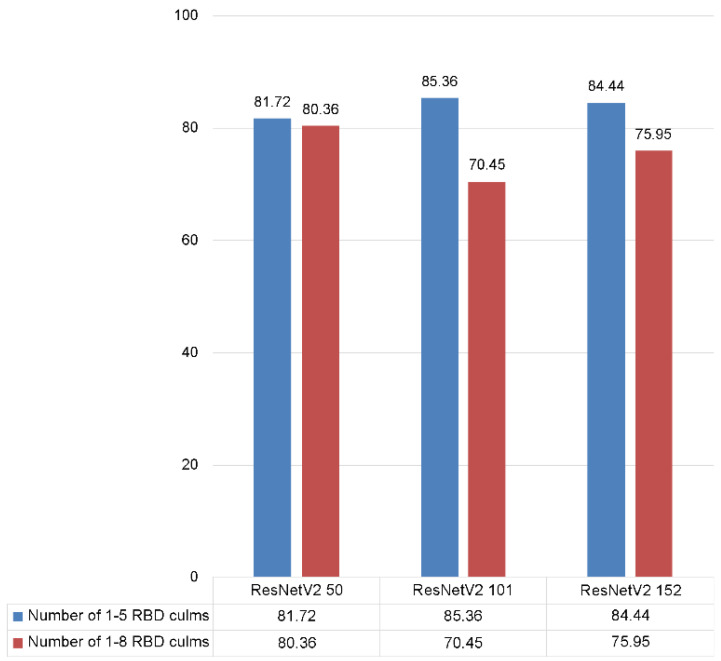
The comparison of classification performance for RBD infected culm data.

**Table 1 sensors-23-00032-t001:** A comparison of object detection models based on Pascal VOC 2007 dataset.

Model	mAP	FPS
Faster R-CNN (ResNetV2 101) [23]	74.63	7
FPN (ResNetV2 101) [24,25]	75.89	6
Bi-FPN (ResNetV2 101) [26,27]	76.04	23
YOLOv2 (DarkNet 53) [28]	78.60	41
YOLOv3 (DarkNet 53) [29]	80.15	36

**Table 2 sensors-23-00032-t002:** The data collected to select the drone’s shooting height and angle.

Taking Height (m)	Taking Downward Angle (°)				Total
	0	5	10	15	
1	126	126	128	126	506
2	126	126	128	126	506
3	126	124	126	127	503
4	126	124	125	124	499
Total	504	500	507	503	2014

**Table 3 sensors-23-00032-t003:** The number of data collected for the training detection and classification model.

Collection Period	Area	Number of Images
7/2020~8/2020	Jeolla-do	100
7/2021~8/2021	Jeolla-do	7548
	Gyeongsang-do	11,095
	Chungcheong-do	1771
Total		20,514

**Table 4 sensors-23-00032-t004:** The data for classification of RBD infected culms.

Number of RBD Infected Culms	Train	Validation	Test	Total
1	3258	838	94	4190
2	7087	1839	180	9106
3	7655	1870	176	9701
4	5252	1265	112	6629
5	1282	326	70	1678
6	281	71	20	372
7	92	18	17	127
8	16	2	5	23
Total	24,923	6229	674	31,826

**Table 5 sensors-23-00032-t005:** YOLOv3 model training parameters.

Parameter Name	Parameter Value
Batch size	5
Total epochs	120
Batch normalization	True
Batch normalization decay	0.99
Weight decay	5 × $10^{-4}$
Learning rate	1 × $10^{-4}$
Cosine learning rate decay	True
Non-maximum suppression threshold	0.45
Pre-train weight	ImageNet

**Table 6 sensors-23-00032-t006:** YOLOv3 model performance results for culm area detection of RBD.

Network	Warm Up	Label Smoothing	mAP		
			Train	Validation	Test
DarkNet53	False	False	95.09%	80.75%	81.33%
	False	True	95.28%	91.36%	89.22%
	True	False	91.64%	89.95%	88.13%
	True	True	97.84%	92.68%	90.49%

**Table 7 sensors-23-00032-t007:** The deep residual network training parameters.

Parameter	Method
Image size	224 × 224 × 3
Batch size	16
Total epochs	300
Learning rate	0.001
Weight decay	0.00004
Batch normalization	True
Optimizer	Adam
Loss function	Focal loss (α = 4.0, γ = 2.0)
Weight initialization	Xavier initialization
Pre-train weight	ImageNet
Network	ResNetV2(50/101/152)

**Table 8 sensors-23-00032-t008:** The comparison of culm classification performance by deep residual network depth.

Depth	50 Layers	101 Layers	152 Layers
Accuracy	80.36%	70.45%	75.95%

**Table 9 sensors-23-00032-t009:** The number of culm classification performance results for 50 layers of deep residual network.

Number of RBD Infected Culms	Test	True (Pred)	False (Pred)	Accuracy
1	94	89	5	94.68%
2	180	168	12	93.33%
3	176	154	22	87.50%
4	112	99	13	88.39%
5	70	59	11	84.28%
6	20	14	6	70.00%
7	17	11	6	64.70%
8	5	3	2	60.00%
Total	674	597	77	80.36%

**Table 10 sensors-23-00032-t010:** The reconstructed number of RBD infected culm classification training and evaluation datasets.

Number of RBD Infected Culms	Original	Train	Validation	Test
1	4190	3258	838	94
2	9106	7087	1839	180
3	9701	7655	1870	176
4	6629	5252	1265	112
5	1678	1282	326	70
Total	31,304	24,534	6139	632

**Table 11 sensors-23-00032-t011:** The comparison of culm classification performance by deep residual network depth.

Depth	50 Layers	101 Layers	152 Layers
Accuracy	81.72%	85.36%	84.44%

**Table 12 sensors-23-00032-t012:** Culm number classification performance results for deep residual network 101.

Number of RBD Infected Culms	Test	True (Pred)	False (Pred)	Accuracy
1	94	85	9	90.42%
2	180	152	28	84.44%
3	176	153	23	86.93%
4	112	92	20	82.14%
5	70	58	12	82.85%
Total	632	540	92	85.36%

## Data Availability

The datasets generated and/or analyzed during the current study are not publicly available due lack of obtaining an online database but are available from the corresponding author on reasonable request.

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
