# Peer review of "Automated Detection of Rice Bakanae Disease via Drone Imagery"

_sensors, 2022, doi:10.3390/s23010032_

Round 1

Reviewer 1 Report

This paper proposes a system for the forecasting and automated inspection of rice Bakanae disease (RBD) infection rates via drone imagery. Before this manuscript is published, I have some concerns.

Line 60-61 recommends adding references, and the format of references at the end of the article needs to be adjusted.

In section 2.3.1, when selecting the object detection model, the YOLOV3 model was selected based on the open VOC dataset. Rather, it is more appropriate to use the results of training the model on your own dataset to select the model.

Line 281: The size of the dataset in the article is 555*555, but it needs to be adjusted to 224*224 when entering the model, so what method is used to adjust it? Why not crop to a 224*224 image at first.

Line 296: ‘Inaccurate labeling will decrease classification accuracy.’ Whether it can provide a standard to judge whether the label is inaccurate and how much accuracy is reduced.

In 3.2.1, the author mentioned that the vertical height is set as 1, 2, 3 and 4m, the basis of which can be explained a little. In addition, a more refined gradient above and below 3m can be selected for the experiment.

Line 399-400: Is the data division an artificial division or a random division? Please explain in detail.

Table 15, the train, validation, and test sets are not divided according to the 7:1:2 in the text.

In addition, too many figures and tables in this paper. Too many formal errors in this paper. The part 3 is logic confusion, and the core research content needs to be condensed. The article is particularly unreadable. It is suggested that the part 2 and 3 be rewritten.

Reviewer 2 Report

Authors have worked automated inspection of Rice Bakanae Disease (RBD) infection rates via drone imagery. They employed YOLOv3 and RestNETV2 101 models for detection and classification of the infected culm numbers, respectively.

Related work/ literature survey section should be included. In this section, authors can include the existing work which was done on RBD specifically using CNN algorithms for classification and detection.

In line no 16, Each letter of abbreviation should be in capital letter. rice Bakanae disease (RBD) -> . Rice Bakanae Disease.

In line 25, m should be written with full name i.e. meters for improving readability.

Resolution of figure 4 needs to be enhance. If it is drawn from other source, reference must be given.

Format needs to be done on page no 6, specifically for figure 6. It is repeated.

Line no. 193-200, they are center aligned. It should be as per format.

Authors have considered YOLOv3 as an object detection model. However, there are other state-of-the-art object detection models are designed and published recently with better performance compare to Yolov3. Moreover, Yolov3 struggle small objects and authors have attempted to detect object from arial images. Justify: Why YOLOv3 and NOT recent object detector? Table 1 shows older object detection comparison and not with recent object detector.

Why author have used two stage approach? Actually, classification is a part of object detection. With the use of state-of-the-art object detection model, accuracy can be improved in less time compared to the presented approach.

Line 315, formatting issues.

Section 3.6. Camera Calibration is elaborated in much detail. It is advised to keep specific information only.

Line 571-575, formatting issues.

Figure 12 and figure 24 represents same images. It should be avoided.

“Figure 25” names are given with A,B multiple times. It creates confusion. Example images should be precise. It is advised to keep specific information only.

Above comment: Same for figure 26

Line 676, formatting issues.

Table 14 should be derived from the code itself. It should not be manually added in table.

Line 692-695, formatting issues.

Table 16, It should be 84.44% in 152 layers, NOT 84,44%.

General Remarks:

Extensive formatting of manuscript is required.

It seems that authors have elaborated a lot including figures. It should be avoided.

Round 2

Reviewer 1 Report

I have no further comment

Reviewer 2 Report

Equations can be typed as per the manuscript font type requirement. 

Authors have addressed the comments.